# Modeling p*K*_a_ of the Brønsted Bases as an Approach to the Gibbs Energy of the Proton in Acetonitrile

**DOI:** 10.3390/ijms231810576

**Published:** 2022-09-12

**Authors:** Zoran Glasovac, Borislav Kovačević

**Affiliations:** 1Division of Organic Chemistry and Biochemistry, Ruđer Bošković Institute, Bijenička c. 54, HR-10000 Zagreb, Croatia; 2Division of Physical Chemistry, Ruđer Bošković Institute, Bijenička c. 54, HR-10000 Zagreb, Croatia

**Keywords:** basicity, p*K*_a_, solvation, quantum chemical calculations, Gibbs energy of a proton

## Abstract

A simple but efficient computational approach to calculate p*K*_a_ in acetonitrile for a set of phosphorus, nitrogen, and carbon bases was established. A linear function that describes relations between the calculated Δ*G*’_a.sol_(BH^+^) and p*K*_a_ values was determined for each group of bases. The best model was obtained through the variations in the basis set, in the level of theory (density functionals or MP2), and in the continuum solvation model (IPCM, CPCM, or SMD). The combination of the IPCM/B3LYP/6-311+G(d,p) solvation approach with MP2/6-311+G(2df,p)//B3LYP/6-31G(d) gas-phase energies provided very good results for all three groups of bases with R^2^ values close to or above 0.99. Interestingly, the slopes and the intercepts of the obtained linear functions showed significant deviations from the theoretical values. We made a linear plot utilizing all the conducted calculations and all the structural variations and employed methods to prove the systematic nature of the intercept/slope dependence. The interpolation of the intercept to the ideal slope value enabled us to determine the Gibbs energy of the proton in acetonitrile, which amounted to −258.8 kcal mol^−1^. The obtained value was in excellent agreement with previously published results.

## 1. Introduction

Considering that the deprotonation of the reactant is a crucial step in many organic reactions, organic (super)bases have wide applications as catalysts in organic synthesis [1,2]. Besides well-known guanidine derivatives, Schwesinger’s iminophosphoranes (phosphazenes) and vinamidines are probably the best-known examples [3,4,5,6,7,8,9]. During the last ten years, a significant amount of work has been conducted on the design and synthesis of triguanides [10] and their relatives termed BIG-bases [11], cyclopropenimines [12,13,14], and proton sponges based on phosphazene and cyclopropenimine subunits [15,16,17,18]. A common feature of all aforementioned (super)bases is the nitrogen atom that serves as a proton acceptor, and they are commonly termed nitrogen superbases (hereafter termed N-bases).

Despite recognizing phosphines as Brønsted bases, they are better known as ligands for transition metals [19]. Since the 1990s, many novel phosphines have been developed with a strong indication of their superbasic properties [20,21]. Perhaps, the most known phosphine superbases are proazaphosphatranes (Verkade bases) [22,23,24,25,26,27,28], tetramethylguanidinophosphines [29,30,31,32], and trisphosphazenophosphines (PP3) [33,34]. Tetramethylguanidinophosphines with measured ^THF^p*K*_a_ above 24 and ^ACN^p*K*_a_ above 29 [32,34,35,36,37] are comparable or even stronger bases than Schwesinger’s P2 phosphazenes, whereas trisphosphazenophosphines appear to be even stronger than P4 phosphazenes.

Other non-nitrogen superbases that came to attention in the last decade are phosphorous ylides, whose basicity limit is estimated to surpass those of phosphazenes [38]. For the ylides bearing Schwesinger’s P1 phosphazene subunits (P1 ylides), p*K*_a_ of up to 33.5 in THF was measured, and this p*K*_a_ value is comparable to Schwesinger’s P4-phosphazene base [39]. Ylide functionality was also employed in the synthesis of naphthalene-1,8-bisylide, the Proton Sponge^®^ analog with high basicity and weak tendency to form an intramolecular hydrogen bond [40].

Hard work devoted to the design and measurements of novel non-nitrogen superbases has been generally supported by gas-phase basicity (GB) and p*K*_a_ calculations with various success. Unlike nitrogen bases, the standard DFT theoretical models for the GB and p*K*_a_ calculations of phosphines have previously led to significant errors. Utilizing B3LYP/6-311+G(2df,p)//B3LYP/6-31G(d) in combination with PCM HF/6-31G(d) solvation-energy calculations, Kovačević and Maksić [41] estimated ^ACN^p*K*_a_ of phosphazenyl phosphine (PP3H^+^) to ca 50, which appeared to be overestimation by ca 7 p*K*_a_ units as shown later on [34,42]. This apparently originates from the substantial error in the computed GB of the P(N(Me)CH_2_CH_2_)_3_N (Verkade base), which served as a reference for p*K*_a_ calculations. A year later, Koppel, Leito, and coworkers [37] measured the gas-phase basicity of the Verkade superbase and showed that the discrepancy against the B3LYP calculated value was higher than 20 kJ mol^−1^ (>4 kcal mol^−1^). DFT methods have also lead to the underestimation of the calculated basicity of tricyclohexylphosphine (PCy_3_), as demonstrated in a paper of Liu et al. [43]. According to their results, the MP2/6-311+G(2df,p)//B3LYP/6-31+G(d) approach outperformed all tested DFT methods. Nevertheless, the possibility of choosing an inappropriate reference base still remains.

The simplest approach to calculate the p*K*_a_ values of species **A**H (acid) or **B**H^+^ (conjugate acid of base **B**) is based on the thermodynamic cycle [44] and can be expressed by Equation (1) [45].
p*K*_a_(**A**H/**B**H^+^) = Δ*G’*_a,sol_(**A**H/**B**H^+^)/ln(10)*RT* + *G*_sol_(H^+^)/ln(10)*RT*(1)

Equation (1) relies on the definition of Δ*G’*_a,sol_(**A**H/**B**H^+^), which stands for the reduced basicity of base **A^−^** or **B** in solution and equals difference in the Gibbs energies of the neutral and protonated forms of the acid/conjugate base pair. *G*_sol_(H^+^) is the Gibbs energy of the proton in a given solvent; *R* is a general gas constant; and *T* is the temperature.

The separation of the Gibbs energy of deprotonation (Δ*G’*_a,sol_) from the Gibbs energy of the proton (*G*_sol_(H^+^)) in a given solvent **S** allows the experimental data for *G*_sol_(H^+^) to be used if available. Since the second term of the equation is a constant (for a given computational approach), Equation (1) could be used for the prediction of p*K*_a_ even in a solvent for which the solvation energy of the proton is unknown. This is usually performed by creating a linear fit for a series of known bases, which at the same time serve as multiple and significantly more reliable reference points. The second problematic point in p*K*_a_ calculations is the accuracy of the computed solvation energies of the neutral and protonated forms. Widely used continuum solvation models (CSMs) usually suffer from systematic errors, as it was shown by Klamt et al. [46,47], leading to the slope of the linear function that significantly differs from the ideal value, which amounts to 1/ln(10)*RT*. Nevertheless, linear functions that use the method-dependent slope and intercept employed by Perakylla [48] and by us [45,49] proved to be a powerful tool for predicting p*K*_a_s, presumably due to the self-correction of the systematic errors [45]. Similarly, Busch et al. recently described the universal approach to p*K*_a_ values in non-aqueous solvents from experimental data measured in water [50]. They emphasized a need for a scaling approach due to the shortcomings of the SMD solvation model. Besides that, the linear function given in Equation (1) also indicates a possibility for determining *G*_sol_(H^+^) from measured p*K*_a_s. Due to the high importance of *G*_sol_(H^+^) [51], its evaluation from the intercept of such function seems very attractive and was already attempted by Matsui et al. by employing their AKB Scheme [52,53]. The AKB scheme is a mathematically slightly rewritten form of our approach in which the authors define the scaling factors that multiply the ideal slope in their fitting procedure. In line with comments on the systematic errors, Δ*G*_sol_(H^+^) values obtained using the AKB scheme significantly differ with respect to the employed computational method and the structure of the acid/base in question. Therefore, Δ*G*_sol_(H^+^) values obtained through the AKB scheme cannot be considered a good estimate unless highly accurate computational methods with scaling factors of 1 are used.

In this work, we present the results of our linear fitting approach to calculate the p*K*_a_ values of neutral nitrogen, phosphorus, and carbon organic (super)bases. Due to the observed shortcomings of the B3LYP methods mentioned above, we considered that our original approach [45] needs improvement and decided to re-examine various computational methods, emphasizing the p*K*_a_ values of phosphines. The accuracy of the gas-phase basicity calculations was tested by employing several density functionals and the MP2 level of theory. The solvation energies of neutrals and ions were calculated using the IPCM, CPCM, and SMD continuum solvation models. By taking advantage of several computational approaches used in this work, we found that a very good estimate of Δ*G*_sol_(H^+^), which is method and structure independent, may be derived from p*K*_a_s through the “two–fit” approach. The second fit was introduced to overcome the systematic errors in p*K*_a_ calculations inherited from the implicit solvation models [46].

## 2. Results and Discussion

The structures of phosphines with experimentally measured gas-phase basicity (GB) are given in Figure 1.

We shall start our discussion with a brief comment on the derivatives with measured GBs. The experimental GBs of phosphines **1**–**10** (Table 1) were mostly reliable data taken from the NIST table or from the paper by Kaljurand et al. [37,43,54]. Phosphine **8** deserves a separate comment. The authors measured proton affinity PA(**8**) using the Cooks simple kinetic [55] and the bracketing technique [56], assuming similar entropies of protonation (Δ*S*_p_) of the measured and reference bases. However, Δ*S*_p_ of the diamines used for the bracketing measurements was expected to significantly differ from that of phosphine **8**. In these cases, GBs instead of PAs should be used [57,58,59]. Therefore, we recalculated GB(**8**) by taking the GB data of the reference bases and obtained 236 ± 3 kcal mol^−1^ (see Appendix A). This value was used for the initial testing of the computational approaches in the gas phase.

The gas-phase basicity values of ten phosphines (Figure 1) were calculated using seven different computational approaches, and the results were compared with the literature data [54].

The most difficult GB to reproduce was that of Verkade base **10,** since all DFT approaches deviated by 6.2–12.7 kcal mol^−1^. MP2 calculations (models M5 and M6) provided a very good estimation of the GB, and they favorably compared with the value of 258.5 and 258.9 kcal mol^−1^ calculated using the G3B3 and G4 approaches, respectively. These two models also showed the best correlation data, with M5 being slightly closer to the ideal slope and intercept values of 1 and 0, respectively. On the other hand, the M6 model showed very close alignment to a trendline obtained for a set of nitrogen bases (Figure 1). The same comparison derived from the M1 data is given in Appendix A. The result favored our corrected value for the GB of base **8** over the literature data.

## 3. Calculation of p*K*_a_

As mentioned in the Introduction, our approach to p*K*_a_ is based on a simple thermodynamic cycle (Appendix A). The choice of the model and a more detailed mathematical description of the calculation is given in the Appendix A. The Gibbs energies of each species in solution (*G*_sol_(**B**) and *G*_sol_(**B**H^+^)) were obtained by adding the Gibbs energy of solvation (Δ*G*°_s_) to the gas-phase Gibbs energies of the neutral and protonated species. The reduced basicity in solution (Δ*G’*_a,sol_) was then calculated according to Equation (2).
Δ*G’*_a,sol_(**B**H^+^) = *G*_sol_(**B**) − *G*_sol_(**B**H^+^) = *G*_(g)_(**B**) + Δ*G*_(s)_(**B**) − *G*_(g)_(**BH**^+^) − Δ*G*_(s)_(**BH**^+^)(2)

The calculated p*K*_a_ values were then obtained from Δ*G’*_a,sol_(**B**H^+^) by applying a linear function (Equation (3)). The corrections for the change in the standard states of the neutral and protonated species canceled each other, while that of proton remains within the intercept (see Appendix A).
p*K*_a_(**B**H^+^) = *a* × Δ*G’*_a,sol_(**B**H^+^) + *b*(3)
where *a* and *b* are the slope and intercept obtained from the p*K*_a_ vs. Δ*G’*_a,sol_(**B**H^+^) linear regression.

The Gibbs energies for a series of phosphorus bases (Figure 2) were calculated using several DFT and MP2 approaches in combination with IPCM, CPCM, and SMD implicit solvation models (see Computational details). The ^ACN^p*K*_a_(**B**H^+^) obtained using M6 gas-phase calculations are given in Table 2. The results obtained using other models are given in Appendix A.

Of the employed computational approaches, CPCM//M6 and IPCM//M6 showed the best overall agreement with the experimental results (Table 2 and Appendix A). In both cases, the correlation coefficient, R^2^, was above 0.99. The mean unsigned error (MUE) and mean square root (RMS) on p*K*_a_ were similar, being 1.0 or lower. The most pronounced differences were the slopes of the trendline, which amounted to 0.637 (IPCM//M6) and 0.690 (CPCM//M6), with both being significantly lower than the theoretical value of 0.733. The largest absolute deviations obtained using these approaches for base **37** were around 3 p*K*_a_ units above the experimental value. The SMD1//M6 approach provided an almost ideal slope value, while the correlation coefficient significantly dropped to 0.977. The reason for that lay in the large absolute error calculated for the most basic derivatives going above 4.5 p*K*_a_ units for base **35,** while several bases deviated by 3 or more p*K*_a_ units from the experimental value. Since all these three methods use the geometries and GBs calculated at the same level of theory, the quality of the correlation differences could only be attributed to the solvation model. We could note that the usage of the CPCM and IPCM solvation model decreased |Δ(p*K*_a_(**35**H^+^)| to 2.0 and 0.7, respectively, which was a significant improvement over the SMD1//M6 result. We also performed p*K*_a_ calculations using several other computational schemes, and the results are given in Appendix A.

Interestingly, model SMD2//M3 provided very good estimates of p*K*_a_ with a relatively high R^2^ value and an MUE below 1.0 p*K*_a_ units. Bases **8**, **10,** and **14** deviated from the trendline by more than 3 units, which was presumably due to the gas-phase calculations. In this case, the MP2 single-point calculation did not improve the correlation. A comparison of IPCM//M5 and IPCM//M6 results revealed a minor but negative effect of the M06-2X/cc-pVDZ optimized geometries on the correlation quality. It is worth mentioning here that the relative trend of the p*K*_a_ values of diphosphines **29**–**31** was only correctly interpreted using IPCM//M5. Among the other methods, SMD2//M3 provided very good estimates of p*K*_a_, but their relative trend was incorrect. All three methods based on M6 geometries underestimated p*K*_a_(**31**H^+^) and positioned it below **30** on the basicity scale.

The selected computational models were additionally tested against the set of nitrogen bases (Figure 2). The resulting trendlines were almost parallel to those obtained for phosphines. For the nitrogen bases, IPCM//M6 performed significantly better than CPCM//M6, giving an R^2^ value of 0.992, while the MUE and RMS amounted to 0.5 and 0.7, respectively. The results indicated a significant improvement over the IPCM//M1 approach recommended earlier [45]. The corresponding values for the CPCM//M6 were R^2^ = 0.977, MUE = 0.8, and RMS = 1.1, which made it less general than IPCM//M6. The SMD1//M6 approach provided results between the other two M6-based models, with the slope being the most similar to that obtained for phosphorus bases among all employed computational models. Again, SMD2//M3 was very good but not the best model, with quite a large difference in the slopes of the linear fits.

After establishing the best model for nitrogen and phosphorus bases, we tested it against a set of phosphorus ylides (Figure 3) whose ^ACN^p*K*_a_ values were published by Leito and coworkers [39,68]. Most of these bases were measured in THF, and their p*K*_a_ values in acetonitrile were estimated by linear regression. The calculated GBs, solvation energies, and ^ACN^p*K*_a_s are given in the Appendix A. The correlation between the computed and experimental values is given in Figure 3.

Besides five significant outliers, the calculated Δ*G’*_a.sol_ values correlated well against experimental ^ACN^p*K*_a_s. The correlation coefficient (R^2^) of 0.982 was slightly lower than those for P and N bases but could still be used with good confidence, as evidenced by the MUE of 0.5 and the RMS of 0.6. The relatively large slope was somewhat surprising, considering that we used IPCM as the solvation model. The slope deviation from the ideal value appeared to follow the hardness of the bases, being the largest for the nitrogen bases and lowest for the phosphorus ylides (C bases). The employed model did not give satisfactory results for the systems bearing three 2,6-disubstituted phenyl rings attached to the phosphorus (bases **Y6**, **Y28,** and **Y29**) and the bases containing phenyl and methyl substituents directly attached to the deprotonation center (**Y28** and **Y29**). Somewhat surprising was a relatively large deviation of base **Y27**—structurally very similar to phosphazenes as well as trismesityl derivative **Y6**. The origin of these discrepancies is currently under investigation.

## 4. Estimation of Δ*G*_sol_(H^+^)

Before discussing the estimation of *G*_sol_(H^+^), we need to make a clear semantic distinction between the literature value and the one obtained from the set of p*K*_a_ calculations. Hereafter, we term them *G*_lit_(H^+^) and *G*_sol,calc_(H^+^), respectively.

As already mentioned, the linear correlation approach used in this work (Appendix A) should allow *G*_sol,calc_(H^+^) to be estimated from the intercept. To test the validity of this idea and the influence of systematic errors, we calculated *G*_sol,calc_(H^+^) for each employed method irrespective of its R^2^. All methods provided sufficiently good correlation coefficients (R^2^ > 0.95) to justify such a decision. Due to the systematic deviation from the ideal slope present in all correlations, Equation (3) was rewritten in the following form.
2.303RT × p*K*_a_(**B**H^+^) = *m* × Δ*G’*_a,sol_(**B**H^+^) + *n*(4)
where *m* and *n* are the coefficients obtained from the correlation parameters and the intercept *n* includes *G*_sol_(H^+^) (Equation (4)). Here, we should emphasize that our parameters *m* and *n* correspond to the *k* and *C*_0_ coefficients used in AKB scheme (Equation (4) in Ref. [52]).

The calculated *m* and *n* values, along with the calculated Gibbs energy of the proton in acetonitrile (*G*_sol,calc_(H^+^)) are presented in Table 3.

To estimate whether the same scaling factor could be used for the slope and intercept, we calculated the value of *n’* (Equation (5)) and compared it to *m*.
*n’* = *n*/*G*_lit_(H^+^)(5)

The literature value of *G*_lit_(H^+^) = −257.3 kcal mol^−1^ was obtained using Equation (6).
*G*_lit_(H^+^) = *G*°_gas_(H^+^) + Δ*G°*_s_(H^+^) + *G*°^→^*(H^+^)(6)
where *G*°_gas_(H^+^) = −6.28 kcal mol^−1^, the literature value for Δ*G*°_s_(H^+^) is −252.9 [49] and the correction for the change in the standard state (*G*°^→^*(H^+^)) is 1.89 kcal mol^−1^.

Calculated parameters *m* and *n’* showed mutual linear dependence (Figure 4a). Assuming *m* = *n’*, we obtained *G*_sol,calc_(H^+^) ranging from −244 to −267 kcal mol^−1^ (Table 3), which corresponded to the results obtained using the AKB scheme [52,53]. One could notice that the linearity was kept even across the three different types of bases. The linear function with a non-zero intercept indicated that the intercept most likely contained two parts; one of them was not related to the Gibbs energy of the proton and should not be scaled with the same scaling factor. This issue was eliminated by modifying the second linear plot, as shown in Figure 4b.

To avoid the usage of any reference point (any published *G*_lit_(H^+^) in this case), we correlated the intercept *n* against *m* (Figure 4b), and the value of *G*_sol,calc_(H^+^) was determined for *m* = 1 (the hypothetical, optimal result without systematic errors, *G°*_sol,opt_(H^+^)), which amounted to −258.8 kcal mol^−1^. Our estimate for the Gibbs solvation energy of the proton in acetonitrile (Δ*G°*_s_(H^+^)) of −254.4 kcal mol^−1^ was then calculated using Equation (6). The value fell in between those obtained by Carvalho and Pliego [69] and Rossini and Knapp [70]. The result was also very close to the standard Gibbs energy of solvation of a proton published by Fawcett [71]. We would like to emphasize here that the estimate was independent of the type of bases, whereas the selection of the computational model had a minor effect. The results showed that a very good estimate of Δ*G*_sol_(H^+^) could be derived from experimental p*K*_a_ despite the systematic errors present in the continuum solvation model approaches.

## 5. Computational Details

All calculations were performed using the Gaussian09 program package [72]. The geometry optimizations and the single-point energy refinements were carried out using either B3LYP [73,74,75,76], M06-2X [77] density functionals, B2-PYLP [78] hybrid functional with perturbative correction or MP2 [79,80] calculations as defined in a list of the models. The minima on the Born–Oppenheimer potential energy surface were verified using a vibrational analysis. The solvation-energy contributions were calculated using the IPCM [81], SMD [82] or CPCM [83,84] continuum solvation model assuming the dielectric constant (ε) of acetonitrile to be either a default implemented value (35.688 for CPCM and SMD) or 36.64 (IPCM). Additionally, the non-standard parameters for IPCM calculations were used for the sake of consistency with our previous results on nitrogen bases, as well as to ensure good convergence [45]. The isodensity parameter of the molecular cavities in solution embracing the solute molecule was 4 × 10^−4^, and the angular integration weights over polar coordinates *θ* and *φ* were set to 100 and 20, respectively. The geometries of the optimized structures were visualized using MOLDEN5.9 [85,86].


*List of models:*


M1 = B3LYP/6-311+G(2df,p)//B3LYP/6-31G(d),M2 = B3LYP(GD3)/6-311+G(2df,p)//B3LYP(GD3)/6-31+G(d),M3 = M06-2X/aug-cc-pvtz//M06-2X/cc- pVDZM4 = M06-2X(GD3)/6-311++G(3df,2pd)//M06-2X/cc- pVDZ,M5 = MP2(fc)/6-311+G(2df,p)//M06-2X/cc-pVDZ.M6 = MP2(fc)/6-311+G(2df,p)//B3LYP/6-31G(d).M7 = B2PLYP/6-311+G(2df,p)//B3LYP/6-31G(d)IPCM = IPCM(ACN)/B3LYP/6-311+G(d,p)CPCM = CPCM(ACN)/B3LYP/6-31G(d)SMD1 = SMD(ACN)/M06-2X/6-31G(d)SMD2 = SMD(ACN)/M06-2X/6-31+G(d,p)

## 6. Conclusions

Based on all examined correlations, IPCM//M6 and CPCM//M6 represented powerful approaches for the prediction of p*K*_a_ in acetonitrile regardless of the type of bases, and the recommended equations for the IPCM//M6 model are as shown below.
N-Bases: p*K*_a_(**B**H^+^) = 0.601 × Δ*G’*_a,sol_(**B**H^+^) − 146.8
P-Bases: p*K*_a_(**B**H^+^) = 0.636 × Δ*G’*_a,sol_(**B**H^+^) − 158.5
C-Bases: p*K*_a_(**B**H^+^) = 0.705 × Δ*G’*_a,sol_(**B**H^+^) − 178.3

The SMD solvation model gave the smallest systematic deviation of the slope, which indicated the best estimate of the changes in the solvation energies. However, the scattering of the correlated data was more pronounced than for the other two approaches, and as such it was less reliable. Due to significant data scattering, we do not recommend using B3LYP calculations for either phosphines or ylides. M06-2X could be used in combination with correlation-consistent basis sets for phosphines, although a larger error could be expected for highly crowded systems such as tricyclohexylphosphine (**8**) and tris(*t*-butyl)phosphine (**14**).

Further, we showed that the approach for estimating the Gibbs energy of solvation of a proton could be applied as long as the reliable p*K*_a_ values were measured. The approach compensated for the systematic errors inherently present in CSM calculations and provided values in a very good agreement with the literature ones. Malloum et al. were absolutely right in their statement: “... the accurate computation of the p*K*_a_ of organic compounds in a given solvent is directly dependent on the accurate determination of the solvation free energy of the proton in that solvent.” [51]. However, the accurate determination of solvation-free energies is not a simple task. Expectedly, we obtained different systematic errors of the same method for the different classes of compounds. Within our approach, self-corrections of the systematic errors allowed us to obtain a very good estimate of the solvation-free energy of the proton in acetonitrile. The plots given in Figure 4 indicate that the approach was applicable regardless of whether we considered one or more classes of the organic bases and regardless of the employed computational models.

## Data Availability

The data are available in this publication and Appendix A.

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
