# Peer review of "Modeling pKa of the Brønsted Bases as an Approach to the Gibbs Energy of the Proton in Acetonitrile"

_ijms, 2022, doi:10.3390/ijms231810576_

Round 1
Reviewer 1 Report
Comment to the Author
In this manuscript, the authors presented a simple computational approach to calculate pKa in acetonitrile for a set of phosphorus, nitrogen, and carbon bases. Combination of the IPCM/B3LYP/6-311+G(d,p) solvation approach with MP2/6-311+G(2df,p)//B3LYP/6-31G(d) gas-phase energies provides very good results for the base groups under consideration. The theoretical Gibbs energy of a proton in acetonitrile, equal to -258.8 kcal/mol, is in excellent agreement with previously published results. The work is of interest to readers of IJMS. However, the manuscript needs to take into account the comments below before I can recommend it for publication.
An important issue needs to be clarified.
The following paper should be cited and discussed in an improved version of the manuscript:
Michael Busch, Ernst Ahlberg, Elisabet Ahlberg, and Kari Laasonen, “How to Predict the pKa of Any Compound in Any Solvent”, ACS Omega 2022, 7, 17369-17383; DOI: 10.1021/acsomega.2c01393
Minor issues.
1. Page 1. Abbreviations P1 and GB must be defined.
2. Page 2. The authors should clearly state that the problem of the accuracy of the theoretical solvation energies in an aprotic solvent is being discussed.
3. Fig. 3. Y6, Y26, Y27, Y27, and Y29 bases cannot be described by linear correlation. It is better not to show them in the Figure.
Author Response
The following paper should be cited and discussed in an improved version of the manuscript: Michael Busch, Ernst Ahlberg, Elisabet Ahlberg, and Kari Laasonen, “How to Predict the pKa of Any Compound in Any Solvent”, ACS Omega 2022, 7, 17369-17383; DOI: 10.1021/acsomega.2c01393
Author reply: The paper is cited in the introduction section.
Minor issues.
- Page 1. Abbreviations P1 and GB must be defined.
Author reply: The abbreviation GB is defined, while the part of the sentence with P1 is rephrased to clearly refer to P1 phosphazene subunits.
- Page 2. The authors should clearly state that the problem of the accuracy of the theoretical solvation energies in an aprotic solvent is being discussed.
Author reply: We corrected the sentence in Introduction section to emphasize that the problem of accuracy relates to the theoretical (computed) solvation energies. Necessity to introduce corrective factors for the compensation of systematic errors in the computed solvation energies in aprotic solvents is also described in the above paper. According to the reviewer's suggestion we added additional sentence to point this out.
- Fig. 3. Y6, Y26, Y27, Y27, and Y29 bases cannot be described by linear correlation. It is better not to show them in the Figure.
Author reply: Figure 3 is changed to exclude prominent outliers.
Reviewer 2 Report
The paper by Glasovac and Kovacevic examines the performance of different computational approaches in modeling the pKa values of series of bases in acetonitrile. Then the Gibbs free energy of H+ in the solvent was estimated based on the linear regression model. The paper builds upon previous work of the authors [27] and the works by Matsui.
The linear relationships established in this work may be possibly useful and the result of the procedure estimating free energy of a proton is interesting. However, the paper needs major improvement in presentation before it can be accepted.
Here are several issues to be considered.
1. There are formatting problems in the description of pKa calculations. The "delta" symbols are missing thorought the paper. All symbols appearing in equations need to be explained.
The presentation in ref. [27] was much clearer and I think that the paper will benefit if the scheme of the thermodynamic cycle is moved to the main paper.
2. Abbrevations IPCM, CPCM and SMD should be explained. Default value of dielectric constant of acetonitrile used in CPCM and SMD models should be given (line 276).
3. Labelling in Fig. 2 is too small.
4. I do not understand what authors wanted to say in lines 247-8: "The calculated parameters m and n' seem similar at first sight; however, their mutual dependence is clearly a linear function".
Why the surprise ("however")? Consider the case when m and n' are equal - then their dependence is a perfect straight line.
5. Table S3. Notation is inconsistent (E_scf vs. E_el) and the energies seem to be in a.u. not in kcal/mol.
6. References need a careful check. "[Error! Bookmark not defined.]" appear in several places in the main paper. Journal name is missing in ref. 32b. References at the end of supplementary information need to be supplied.
Author Response
- There are formatting problems in the description of pKa calculations. The "delta" symbols are missing thorought the paper. All symbols appearing in equations need to be explained.
The presentation in Ref. [27] was much clearer and I think that the paper will benefit if the scheme of the thermodynamic cycle is moved to the main paper.
Author reply: All "delta" symbols are corrected by unicode letters, which we do not expect to be wrongfully converted during conversion to PDF format. Thermodynamic cycle (TC) and the mathematical reasoning that leads to equations (2) and (3) are already published in a number of papers, including our previous paper (Ref. 27). In order to reduce the length of the manuscript and focus the discussion on new results, we transferred the methodology of the pKa calculations to Suppl. Materials and provided the reference of our previous paper. In our opinion, moving the TC to the main text would not increase the understanding of the methodology unless most of the equations are also moved, which, together with the explanation of the symbols, would increase the length of the manuscript by at least 1 page. A text describing pKa calculations in Suppl. Mat is revised, and errors and duplications were removed. Additionally, symbols are more thoroughly explained.
- Abbrevations IPCM, CPCM and SMD should be explained. Default value of dielectric constant of acetonitrile used in CPCM and SMD models should be given (line 276).
Author reply: The full names of the solvent models based on the polarizable continuum model (IPCM, SMD, and CPCM) are given in computational details. The abbreviations are coming from the Gaussian09 options within the SCRF keywords, and it is expected to be understandable even to beginners in computational chemistry. The default values of EPS(ACN) used for CPCM and SMD calculations are given in the text.
- Labelling in Fig. 2 is too small.
Author reply: The new version of Figure 2 with significantly larger labels, thicker lines, and larger data markers is introduced.
- I do not understand what authors wanted to say in lines 247-8: "The calculated parameters m and n' seem similar at first sight; however, their mutual dependence is clearly a linear function".
Why the surprise ("however")? Consider the case when m and n' are equal - then their dependence is a perfectly straight line.
Author reply: Two sentences related to the problem statement are rephrased to make it simpler.
- Table S3. Notation is inconsistent (E_scf vs. E_el) and the energies seem to be in a.u. not in kcal/mol.
Author reply: All Tables in Supplementary Materials are checked, and the pointed inconsistency and some additional typos are corrected.
- References need a careful check. "[Error! Bookmark not defined.]" appear in several places in the main paper. Journal name is missing in Ref. 32b. References at the end of supplementary information need to be supplied.
Author reply: Errors in cross-references are corrected
The journal name in Ref. 32b is "J" and is not missing. It is just difficult to see it.
References in Suppl. Materials are added and formatted appropriately.
Reviewer 3 Report
This is a solid (if unspectacular) study of computational methodology for pKa prediction in nonaqueous solvents. The methodology is solid, and the sort of thing that would be useful for non-expert practitioners. The approach is extremely pragmatic, and the availability of data is used to provide a better prediction (via adjustment of the linear/constant parameters in a linear-least-squares fit) than could be achieved more rigorously.
The main typographical revision I see is that "offliners" should be replaced by "outliers."
The paper could be made stronger by providing an estimate of the sensitivity of the parameters in the final model. For example, by doing cross-validation, one could not only provide a more robust error estimate for these fits, but also an estimate of how much uncertainty there is in the fitting parameters. Given that it is very unlikely there there will be any big surprises in such an analysis given the simplicity and (usual) robustness of linear-least-squares, I would consider this an optional embellishment.
Author Response
The main typographical revision I see is that "offliners" should be replaced by "outliers."
Author reply: The word offliners is replaced with outliers throughout the text.
The paper could be made stronger by providing an estimate of the sensitivity of the parameters in the final model. For example, by doing cross-validation, one could not only provide a more robust error estimate for these fits, but also an estimate of how much uncertainty there is in the fitting parameters. Given that it is very unlikely there there will be any big surprises in such an analysis given the simplicity and (usual) robustness of linear-least-squares, I would consider this an optional embellishment.
Author reply: Cross-validation of the pKa/G'a,sol fits (first plots) could be done, but the reviewer also pointed out that they do not expect any big surprises. Actually, the approach we used here is essentially the same as we used in Ref. 27. The improvement relates to the computational method and not the mathematics behind the correlation. The previously published approach (based on B3LYP calculations) was tested against a number of additional bases in Ref. 27 and our subsequent papers. Agreement between computed and measured pKas even for structurally quite different bases (having flexible substituents capable of forming intramolecular hydrogen bonds) was quite satisfactory, confirming that the approach was not over-correlated nor biased toward bases used as the training set. Since the reviewer offered to consider this suggestion an optional embellishment, we decided not to include potential cross-correlation data in the paper.
The language in the manuscript is corrected and improved utilizing Grammarly.